# Evaluation of Consumer Perception of New Aquaculture Products through Applying Focus Group and Check-All-That-Apply Methodologies

**DOI:** 10.3390/foods13162480

**Published:** 2024-08-07

**Authors:** Palmira Javier-Pisco, Isabel Escriche, Marta Igual, Purificación García-Segovia, María Jesús Pagan

**Affiliations:** Instituto Universitario de Ingeniería de Alimentos-FoodUPV, Universitat Politècnica de València, Camino de Vera s/n, 46022 Valencia, Spain; iescrich@tal.upv.es (I.E.); marigra@upvnet.upv.es (M.I.); pugarse@tal.upv.es (P.G.-S.); jpagan@tal.upv.es (M.J.P.)

**Keywords:** ideas, new products, sea bream, prawn, opinion, consumers

## Abstract

A growing interest in healthy diets has increased demand for fish and seafood, with aquaculture playing a crucial role in meeting this need. Developing new aquaculture products can enhance their commercial value and address consumer demand, but it is unclear which products will be well-received. This study aimed to generate ideas for new products derived from sea bream and prawns, and to gather consumer opinions on these ideas, segmented by gender and age. Two methodologies were used: focus groups and Check-All-That-Apply (CATA). In the focus groups, with two sessions per species and 10 participants each, ideas for aquaculture products were generated and categorized as fresh, dehydrated, fermented, marinated, and canned. The CATA technique, applied to 387 individuals, assessed the acceptability of fresh species, yielding average scores of 6.6 for sea bream and 6.8 for prawns. Sea bream was associated with products like fillets and long-shelf-life loins, while prawns were linked to snacks and toppings. In conclusion, the use of tools like focus groups has shown promising results for developing new aquaculture products. CATA analysis indicated that sea bream should be minimally processed with a long shelf life, and prawns should be processed into dehydrated products. Women preferred traditional products, while men favoured innovative options.

## 1. Introduction

A growing number of consumers are increasingly interested in maintaining a healthy and balanced diet [1], with fish and seafood becoming a primary source of nutrients [2]. As a result, the consumption of fishery products has increased significantly in recent decades, with aquaculture playing a pivotal role in this growth [3]. Aquaculture is an alternative that complements traditional fishing [4]. However, most aquaculture products are marketed as whole fish without any treatment or removal of parts. In many countries, such as Spain, less than 20% of aquaculture products are processed. Consumers demand aquaculture products, and there is a shortage of seafood products in the market [5,6]. For this reason, developing new aquaculture fish products presents an opportunity to enhance the commercial value and profitability of the Mediterranean aquaculture value chain [7]. Therefore, offering consumers new aquaculture products to fulfil their needs is essential. Nevertheless, what specific products should be developed? It remains to be seen whether these products are well-received by consumers. Distinct questions arise during this process.

To develop new products, it is imperative to gain a profound understanding of consumer needs and integrate them into every stage of the product development process [8]. Co-creation involves stakeholders, including consumers, in idea generation; it allows customers to co-build the service experience, exploring their desires and needs, and reducing the risk of not meeting their demands. Furthermore, co-creation utilizes traditional creative techniques (which aim to generate ideas) and projective techniques that interpret unconscious desires through stimuli. This combination of tools can be applied to traditional qualitative methodologies, such as focus groups [9]. These exploratory techniques facilitate the gathering of customer ideas on promising new products. These groups typically comprise six to ten individuals and are conducted at the outset of the product development process [10]. The generation of ideas is paramount in identifying key attributes influencing consumers’ purchasing decisions. This process involves the collection of preliminary qualitative data [11].

A more comprehensive comprehension of the sensory experiences of prospective consumers can facilitate the generation of novel ideas and inspire innovation [12]. In this context, many research techniques permit the study and comprehension of consumer perceptions and behaviors [13]. One method for studying these sensory expectations is to employ current methodologies, such as the “Check-All-That-Apply” (CATA) test. This methodology entails presenting a sample to evaluators along with a predefined list of descriptive terms, and then requesting that they check all that they believe describe the sample [14]. This method is beneficial in the initial product development phase because it can rapidly evaluate numerous samples and circumvent the potential discordance that might arise from responses if intensity scales are used [15]. This test requires the administration of surveys to multiple participants to obtain reliable data from the target population with sufficient statistical robustness [16]. Nevertheless, contemporary tools and technological devices are now facilitating this task. As indicated by Ares and Jaeger [17], terms related to other product characteristics may also be included, contingent on the scope of CATA studies. These include hedonic terms, usage occasions, and product positions. In this context, it is possible to use CATA questions and hedonic ratings to gain insights into consumer preferences. The hedonic question should be placed before the CATA question [18]. The food industry regards sensory analysis and consumer behavior research as some of the most valuable tools in the stages of new product development to ensure the success of innovation in market acceptance among consumers [12]. Studies where gender and age should be considered when evaluating consumer purchasing behavior have been reported [2]. For example, it has been observed that women belong to the segment of fish connoisseurs, while men belong to the segment of uninvolved fish consumers, showing lower consumption levels [19]. Therefore, the objective of this study was to co-create new aquaculture product ideas from sea bream and prawn using two tools: (i) focus groups to generate an initial list of product ideas and (ii) the CATA method to identify the most interesting ideas based on consumer opinions globally and segmented by gender and age.

## 2. Materials and Methods

The study was conducted using a multimethod approach. In the first phase, a focus group tool was used as a qualitative technique, while in the evaluation stage of the new aquaculture products, CATA methodology was employed as a rapid association method.

### 2.1. Co-Creation Idea

#### 2.1.1. Focus Group

Two work sessions were conducted for each species: sea bream (representing aquaculture fish) and prawn (for aquaculture seafood), with 10 participants per session recruited from the database available at Instituto Universitario de Ingeniería de Alimentos-FoodUPV (Universitat Politècnica de València, Valencia, Spain) and the Association of Chefs of the Valencian Community (Valencia, Spain). The inclusion criteria were as follows: participants were required to be adults, responsible for purchasing food, and regular consumers of fish and seafood with a frequency of at least once a week. Participation in the study was voluntary.

The focus group sessions were conducted at the facilities of Instituto Universitario de Ingeniería de Alimentos-FoodUPV (Universitat Politècnica de València, Spain). A moderator and a senior laboratory technician acting as facilitators led the work sessions [8].

The work sessions were conducted in three stages. The first stage began with an online survey on the consumption of aquaculture products. Participants were asked to provide their personal information and data on their frequency of seafood consumption. In the second stage, participants were presented with a series of recently launched food products and were asked to identify their features and keywords. The third stage was the actual focus group session, which began with the participants generating new product ideas based on the perceived need or desire to find them in supermarkets. It was followed by evaluating the participants’ opinions and perceptions of the generated ideas. The duration of each focus group session was 90 min. Subsequently, the participants were invited to collaborate in categorizing the ideas for new products generated according to their production technology (fresh, dehydrated, fermented, marinated, and canned).

#### 2.1.2. Data Evaluation

Three researchers (the moderator, the assistant, and the project coordinator) individually generated a report based on the frequency of mentions of the ideas collected from the participants, which were considered qualitative data for sea bream and prawn [20]. Subsequently, a comparison and discussion of the individual reports were conducted to reach a consensus on the definition, name, or terms of the new aquaculture product ideas deemed valid for both species: sea bream and prawn. Afterward, these ideas were classified according to different processing technologies (fresh, dehydrated, fermented, marinated, and canned). This data evaluation, consisting of independently generating individual reports, discussion, and consensus to obtain a list of different topics, has been commonly applied in other studies [21].

### 2.2. Evaluating Ideas for New Aquaculture Products

The ideas for new aquaculture products generated because of the co-creation phase (as explained before in Section 2.1) were used for five species (sea bream, prawn, sea bass, lobster, and salmon) in the CATA methodology [22].

#### 2.2.1. Online Survey Development

The electronic questionnaire used for this task was developed using the RedJade^®^ Online Survey Tool (Redjade Sensory Solutions, LLC, Martinez, CA, USA), which evaluates each aquaculture species in a randomized order [14]. The electronic questionnaire is shown in the Appendix A and consists of three parts. First, the questionnaire sought to ascertain the acceptability of fresh aquaculture species, with each species represented by royalty-free images [23]. Thus, a nine-point hedonic scale was designed for this purpose. Verbal categories are typically assigned numerical values, with “like extremely” assigned a value of “9” and “dislike extremely” assigned a value of “1” [24]. The second part of the study involved the following: CATA analysis, where the list of new aquaculture products (generated in the focus group) was presented, instructing the respondents as follows: “From the following list of products made with [name of the aquaculture species], select all that you would purchase or consume”. The terms and lists were presented randomly [25]. The third part of the study aimed to gather information about the participants; this section included questions regarding the respondents’ gender, age, place of origin, level of education, employment status, personal circumstances, and economic capacity. All the information provided was anonymized under data protection legislation [26]. This study was approved by the Institutional Ethics Committee (ref. P07_22-06-2022).

#### 2.2.2. Participants’ Recruitment

The recruitment process was conducted via email, with a hyperlink in each message. A total of 387 individuals participated in the study (57% women, 42% men, and 1% non-binary persons) based on the inclusion criterion of consuming aquaculture products at least once a week. The online questionnaire was accessible for 4 weeks.

#### 2.2.3. Statistical Analysis

A multifactorial ANOVA was conducted to assess the statistical significance of the observed differences in the acceptability of aquaculture species (based on the scores obtained from participants’ hedonic scales). Factors included age and gender, as well as other sociodemographic aspects of the participants, such as place of residence, level of education, current status, lifestyle, and grocery budget. In the final model, only the significant factors and their interactions were considered as required for this type of statistical test. The Fisher least significant difference (LSD) procedure was employed for multiple comparisons, with a significance level of α = 0.05. The analysis used Statgraphics Centurion 19 software (Statgraphics Technologies, Inc., The Plains, VA, USA).

The results of the CATA questionnaire were used to calculate Cochran’s Q to determine the existence of significant differences. Correspondence analysis (CA) and agglomerative hierarchical clustering (AHC) were conducted using Euclidean distance and Ward’s method to identify homogeneous groups among species and products. Correspondence factor analysis (CFA) was performed with gender and age as variables. All analyses used XLSTAT 2023.1.4 software (Lumivero, LLC, Denver, CO, USA).

## 3. Results and Discussion

### 3.1. Focus Group

In total, 68 new product ideas were generated. A total of 35 new product ideas were generated for sea bream (Figure 1) and 33 for prawns (Figure 2).

The complete list of generated ideas (*y*-axis) and the percentage of mentions for each idea by participants (*x*-axis) are presented in Figure 1 and Figure 2. The ideas were subsequently grouped into product categories, including fresh, dehydrated, fermented, marinated, and canned. According to Carlucci et al. [2], studies highlight that the growing demand for convenience has driven the introduction of various fish products with different levels of processing.

Upon examination of the results by product category, it was observed that participants generated a greater number of product ideas within the fresh product category for both species: sea bream (Figure 1) and prawn (Figure 2).

Regarding sea bream (Figure 1), the dehydrated product category exhibited the second-highest level of participation in generating ideas. In contrast, the remaining product categories, including those for prawns (Figure 2), exhibited a comparable level of participation. The high rate of mentions of ideas in fresh products may be attributed to the observation by Carlucci et al. [2] that consumers do not receive appropriate seafood preservation methods. Moreover, changes in appearance, presentation, and packaging are essential for new products [27].

In the category of dehydrated products for both species (Figure 1 and Figure 2), mentions of snacks, dehydrated slices, and salted dehydrated products were frequently observed. Prakasan et al. [27] suggested a high demand for dried, seasoned, and convenient-to-use seafood products. Consequently, it is imperative to consider consumer opinions.

In the canned sea bream category (Figure 1), 15% of the mentions referred to options of different flavors, including oriental sauce with vegetables. As indicated by Prakasan et al. [26], to increase the popularity of value-added fishery products worldwide, it is crucial to diversify products with international flavors, including ethnic flavors.

In the marinated products category for sea bream, 20% of the ideas generated were related to flavors and spices. Marinated products appeal to consumers because of their distinctive flavors and textural properties [27].

All ideas generated were evaluated by the researchers (moderator, assistant, and project coordinator), resulting in a list of proposals for new aquaculture products (Table 1) using the following selection criteria: (i) consolidation of products with similar characteristics and (ii) mention frequency. López-Mas et al. [7] used the same criteria, whose textual analysis focuses on keyword frequency.

### 3.2. Acceptability of Aquaculture Species

Table 2 shows the multifactor ANOVA results for the two factors (age and gender) that were significant. This was not the case for the sociodemographic aspects of the participants cited in Section 2.2.3. This table presents the F-ratio, the *p*-value for both factors, and the scores (mean and individual standard error) of the aquaculture species considered (salmon, sea bass, sea bream, prawn, and lobster). The participants who did not specify their gender (1%) were excluded, as there was no representation for all age groups.

Regarding overall ratings, salmon was the most highly rated aquaculture species, with an average score of 7.2. In contrast, lobsters received the lowest rating, with an average score of 6.2. The scores for sea bream, sea bass, and prawns were intermediate (6.6, 6.8, and 6.7, respectively), with no significant differences (*p* > 0.05) among them.

Considering gender, women exhibited higher average scores for salmon, sea bass, sea bream, and prawns than men. These findings are consistent with a previous study that found women more accepting of fish products than men [27]. Furthermore, a considerable proportion of women belong to the consumer segment, demonstrating greater familiarity with fish products, whereas men occupy a less engaged segment [28]. Nevertheless, it is noteworthy that no significant differences were observed between the sexes regarding sea bream and lobsters.

In reference to age, the youngest group (18–24 years old) exhibited the lowest average scores for all aquaculture species compared to the other age groups. As Cardoso et al. [19] observed, the typical profile of fresh fish consumers includes childless couples, adult couples with children, and retired individuals. Moreover, numerous studies have shown that younger individuals perceive fish as more inconvenient than older individuals [2].

The highest average acceptability ratings were observed in the group over 65 for sea bream and sea bass, with significant differences compared to other age ranges. According to the data reported by MAPA [29], retired individuals (over 65) consume the highest amounts of fresh fish, exceeding the market average. As Carlucci et al. [2] observed, some studies have indicated a strong positive correlation between an individual’s attitude toward eating fish and the frequency with which they consume fish.

Regarding aquaculture species, salmon was the most highly rated, with an average score of 7.6 in the age group between 25 and 44 years. However, there were no significant differences compared to other age ranges. The MAPA report [29] indicates that fresh salmon remains a highly accepted product among Spanish consumers willing to purchase more despite price fluctuations. Prawns and lobsters were the most highly rated, with average scores of 7.4 and 7.0, respectively, by the age group of 45 to 54 years, which exhibited significant differences compared to the younger groups (18 to 34 years). According to data from MAPA [29], the consumption of prawns represents over a quarter of the seafood market, reflecting their high acceptability and popularity in the seafood market. No interactions were identified between the factors under investigation (gender and age).

### 3.3. Evaluating Ideas for New Aquaculture Products with CATA Methodology

#### 3.3.1. Cochran’s Q Test

Following an assessment of the suitability of various fresh aquaculture species, the subsequent phase involved examining the proposed new products that could be derived from these species, employing CATA methodology.

Cochran’s Q test enabled evaluation of a selection of new food ideas by participants in each aquaculture species. A series of pairwise comparisons revealed significant differences (*p* < 0.001) between the evaluated products across the species considered in this study, except for “traditional canned stews” and “traditional microwave stews”, which are related to the same recipe. The *p*-values for these comparisons were 0.406 and 0.646, respectively. Table 3 presents the mean scores for the proposed aquaculture product concepts for the five species under consideration.

A significant difference (*p* < 0.001) was observed between fish and seafood regarding fillets, loins, and skinless and boneless halves with long shelf lives. The three types of fish exhibited the highest scores. The snack concept was rated significantly higher for prawns (0.0336, *p* < 0.001). The highest-scoring concepts were salmon and prawns (0.186 and 0.183, respectively), with no significant differences between them. Conversely, the concepts with the lowest scores corresponded to pickled and cured sausage products, where these products with salmon were rated significantly better (*p* < 0.001) than those with lobster, prawn, or sea bass.

#### 3.3.2. Correspondence Analysis

Figure 3 depicts a projective map for the correspondence analysis (CA). This figure illustrates a representation of the five evaluated aquaculture species and proposed new products in the first two dimensions, which collectively explained 89.25% of the variability in the data.

Dimension 1 accounted for 70.24% of this variability, whereas dimension 2 accounted for 19.01%. Dimension F1 distinguishes between lobster, prawn, salmon, sea bream, and sea bass, whereas dimension F2 separates salmon from other aquaculture species. This projection allows the observation of three principal areas: salmon occupying the upper right, lobster and prawn grouped in the lower right, and sea bream and sea bass in the lower left.

Salmon is most commonly associated with value-added products, including those marinated, smoked, or pickled in brine. Conversely, sea bream and sea bass are primarily associated with products such as fillets, loins, and half-butterfly fresh fillets. In accordance with the findings of Ares and Jaeger [18], the spatial proximity of aquaculture species within this framework indicates similarity. Conversely, lobsters are associated with natural preserves, whereas prawns are more closely aligned with toppings and snacks. Seafood (lobster and prawn) is not associated with microwave-ready meals with side dishes or prepared canned products. According to Birtch et al. [30], consumers lack confidence in seafood cooking. Consequently, a ready-to-eat product is suggested.

In the case of sea bream and sea bass, there was no association with any new product proposals related to salmon and seafood. Carlucci et al. [2] observed that most consumers prefer chilled (fresh) fish, with a gradual decline in acceptance of frozen, canned, and smoked/salted fish. Similarly, ready-to-eat fresh fish fillets are highly convenient products because they require no preparation time or effort [2].

The colors depicted in Figure 3 correspond to the AHC method. The color-coded representation indicates the separation of two clusters: green represents C1, and blue corresponds to C2. This shows a more pronounced correlation among prawns, toppings, and snack products.

To gain a deeper understanding of consumer opinions, the proposed new aquaculture products were segmented by gender and age, as illustrated in Figure 4.

The first two dimensions accounted for 59.7% of data variability, with dimension F1 accounting for 40.99% and dimension F2 accounting for 18.71%. Dimension F3 explains 11.72% and is not considered in the analysis. The Dimension F1 classification system differentiates between females and males. All females (F(18–24), F(25–34), F(35–44), F(45–54), F(55–64), and F(>65)) were placed in the left quadrant, whereas most males (except for groups M(45–54) and M(55–64)) were situated in the right quadrant. This projection allows us to observe that females demonstrate a preference for minimally processed products, such as fillets, loins, and half-opened products. Conversely, males are associated with processed products. The younger groups (M_1, M_2, and M_3), situated further to the right along the F1 axis, selected products such as snacks, pâté, pickled items, and cured sausages.

## 4. Conclusions

Applying specific tools for co-creation with consumers has yielded promising results that could drive the development of new aquaculture products featuring sea bream and prawns. During co-creation, focus groups generated a list of new aquaculture product ideas based on various technological processes.

CATA methodology revealed that sea bream is predominantly associated with fresh and minimally processed products, such as fillets, loins, and butterfly cuts, without bones or skin. This highlights the need to develop products with an extended shelf life in these formats. On the other hand, prawn is linked to processed products utilizing dehydration technology, such as snacks for appetizers and toppings.

Additionally, consumer studies have highlighted the importance of considering gender and age segregation in developing products tailored to specific populations. These findings provide valuable guidelines for creating products that better align with the preferences and needs of the target audience. Specifically, women prefer minimally processed aquaculture products, exhibiting more conservative behavior, while men tend to favor processed products, demonstrating a more innovative approach.

## Figures and Tables

**Figure 1 foods-13-02480-f001:**
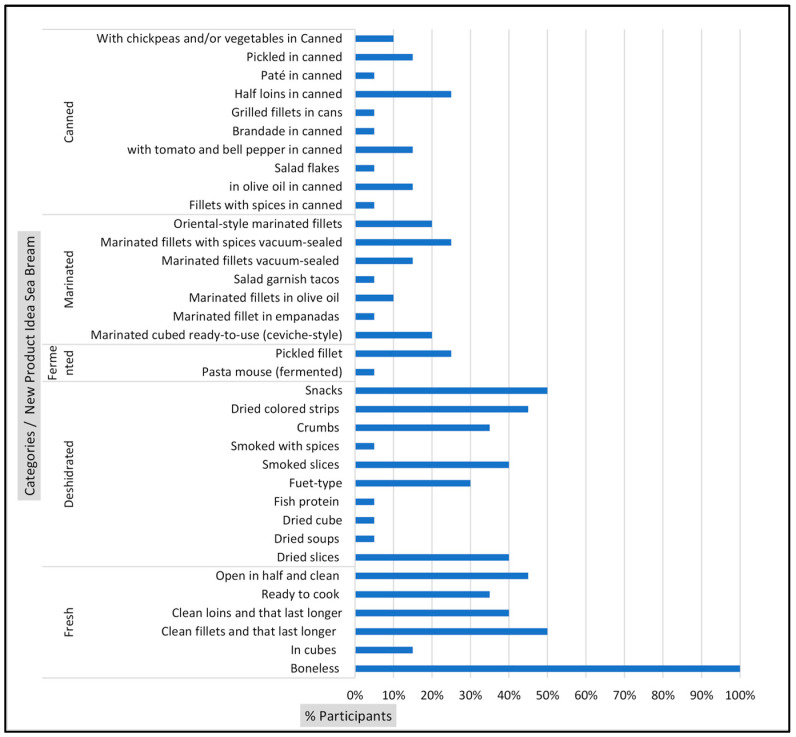
Frequency percentage of ideas for new product ideas for sea bream.

**Figure 2 foods-13-02480-f002:**
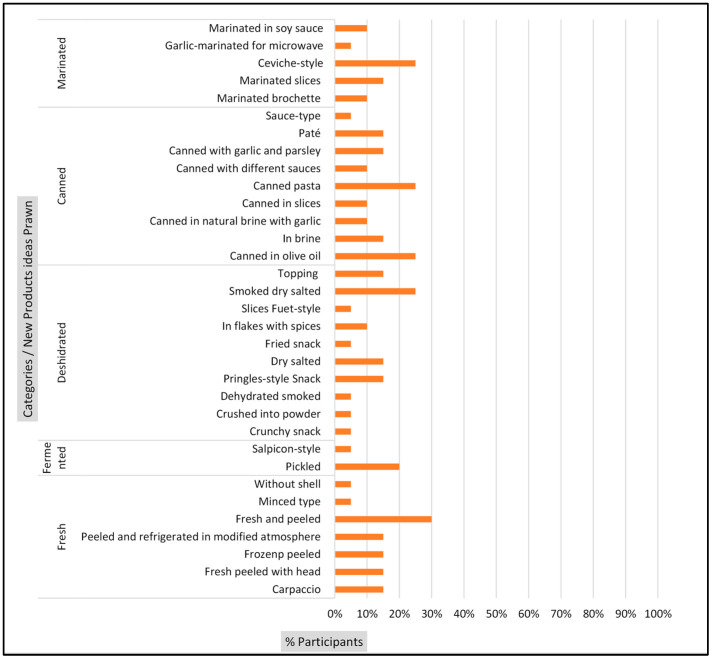
Frequency percentage of ideas for new product ideas for prawn.

**Figure 3 foods-13-02480-f003:**
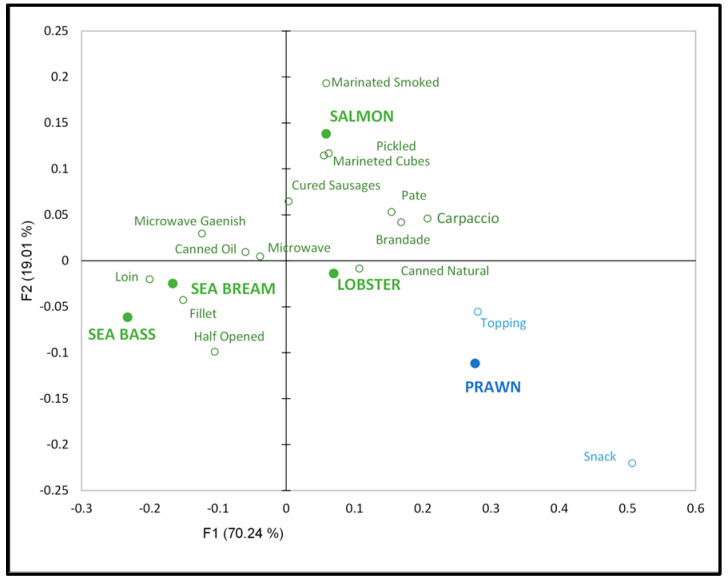
Correspondence analysis Check-All-That-Apply (CATA) data. Note: Different colors identify Agglomerative Hierarchical Clustering (AHC) (green = C1 and blue = C2).

**Figure 4 foods-13-02480-f004:**
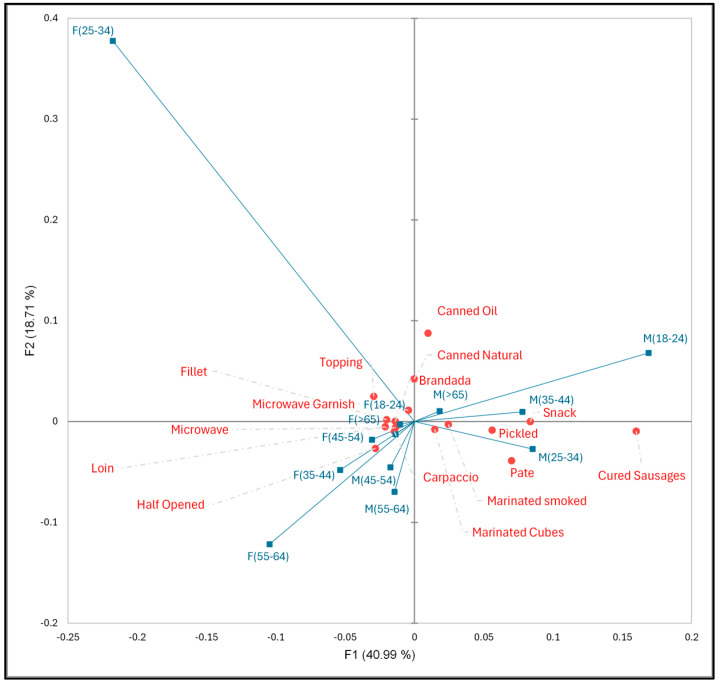
Correspondence analysis CATA data grouped by gender (F: female; M: male) and age (18–24; 25–34; 35–44; 45–54; 55–64; >65).

**Table 1 foods-13-02480-t001:** List of new aquaculture products co-created.

	Code
Fillet: boneless, skinless, long shelf life	Fillet
Loin: boneless, skinless, long shelf life	Loin
Half-Opened: boneless, skinless, long shelf life	Half-Opened
Appetizer: snack	Snack
Topping: dry, salty	Topping
Brandade	Brandade
Canned Natural: preserves	Canned Natural
Canned Olive: preserves with olive oil and tomato sauce	Canned Oil
Pâté	Pâté
Marinated Cubes	Marinated Cubes
Microwave: ready-to-eat	Microwave
Microwave Garnish: ready with garnish	Microwave Garnish
Marinated and smoked	Marinated Smoked
Pickled: in brine	Pickled
Canned Stew: traditional	Canned Stew
Microwave Stews: traditional	Microwave Stew
Carpaccio	Carpaccio
Cured Sausages	Cured Sausages

**Table 2 foods-13-02480-t002:** Multifactorial ANOVA for the acceptance of fresh aquaculture species.

Factors	Salmon		Sea Bass		Sea Bream		Prawn		Lobster	
	Average (SE)	F-Ratio	Average (SE)	F-Ratio	Average (SE)	F-Ratio	Average (SE)	F-Ratio	Average (SE)	F-Ratio
Gender		5.4 *		14.7 ***		1.3 ns		7.2 **		0.0 ns
Male	6.9 (0.2) ^a^		6.4 (0.2) ^a^		6.5 (0.2) ^a^		6.4 (0.2) ^a^		6.3 (0.2) ^a^	
Female	7.5 (0.2) ^b^		7.2 (0.2) ^b^		6.8 (0.2) ^a^		7.1 (0.2) ^b^		6.2 (0.2) ^a^	
Age		1.9 ns		4.7 ***		3.3 **		2.0 ns		3.6 **
18–24	6.6 (0.3) ^a^		5.9 (0.3) ^a^		5.7 (0.2) ^a^		6,1 (0.3) ^a^		5,5 (0.3) ^a^	
25–34	7.6 (0.2) ^b^		6.5 (0.2) ^b^		6.7 (0.2) ^b^		6.7 (0.2) ^a^		6.0 (0.2) ^ab^	
35–44	7.6 (0.3) ^b^		6.8 (0.3) ^bc^		6.7 (0.3) ^b^		6.7 (0.3) ^ab^		6.3 (0.3) ^bc^	
45–54	7.4 (0.3) ^ab^		7.3 (0.2) ^c^		6.8 (0.2) ^b^		7.4 (0.3) ^b^		7.0 (0.3) ^c^	
55–64	7.1 (0.2) ^ab^		6.9 (0.2) ^bc^		6.8 (0.2) ^b^		6.8 (0.2) ^ab^		6.6 (0.2) ^c^	
>65	7.1 (0.5) ^ab^		7.5 (0.4) ^c^		7.1 (0.4) ^b^		6.7 (0.5) ^ab^		5.9 (0.5) ^abc^	

(SE) = individual standard error. Different letters in the same row indicate significant differences at the 95% confidence level obtained by the LSD test. * *p* < 0.05; ** *p* < 0.01; *** *p* < 0.001; ns: not significant.

**Table 3 foods-13-02480-t003:** Cochran’s Q test results for different aquaculture species and proposal of new products.

New Product Proposals	Sea Bream	Lobster	Prawn	Sea Bass	Salmon	*p*-Value
Fillet	0.760 ^b^	0.439 ^a^	0.463 ^a^	0.690 ^b^	0.775 ^b^	***
Loin	0.638 ^b^	0.367 ^a^	0.331 ^a^	0.602 ^b^	0.654 ^b^	***
Half-Opened	0.693 ^c^	0.491 ^a^	0.463 ^a^	0.584 ^b^	0.623 ^bc^	***
Snack	0.103 ^ab^	0.152 ^bc^	0.336 ^d^	0.085 ^a^	0.214 ^c^	***
Topping	0.119 ^a^	0.085 ^a^	0.183 ^b^	0.070 ^a^	0.186 ^b^	***
Brandade	0.121 ^ab^	0.155 ^b^	0.140 ^ab^	0.088 ^a^	0.209 ^c^	***
Canned Natural	0.225 ^ab^	0.243 ^b^	0.230 ^ab^	0.171 ^a^	0.320 ^c^	***
Canned Oil	0.212 ^b^	0.116 ^a^	0.134 ^a^	0.137 ^a^	0.225 ^b^	***
Pâté	0.121 ^ab^	0.119 ^ab^	0.132 ^b^	0.080 ^a^	0.199 ^c^	***
Marinated Cubes	0.230 ^a^	0.191 ^a^	0.204 ^a^	0.186 ^a^	0.401 ^b^	***
Microwave	0.199 ^b^	0.140 ^a^	0.163 ^ab^	0.189 ^ab^	0.269 ^c^	***
Microwave Garnish	0.233 ^b^	0.160 ^a^	0.145 ^a^	0.227 ^b^	0.297 ^c^	***
Marinated Smoked	0.307 ^a^	0.233 ^a^	0.261 ^a^	0.240 ^a^	0.607 ^b^	***
Pickled	0.098 ^bc^	0.070 ^ab^	0.072 ^ab^	0.052 ^a^	0.140 ^c^	***
Canned Stew	0.085 ^a^	0.103 ^a^	0.088 ^a^	0.103 ^a^	0.111 ^a^	ns
Microwave Stew	0.150 ^a^	0.171 ^a^	0.145 ^a^	0.160 ^a^	0.165 ^a^	ns
Carpaccio	0.238 ^ab^	0.302 ^bc^	0.349 ^c^	0.212 ^a^	0.499 ^d^	***
Cured Sausages	0.096 ^bc^	0.049 ^a^	0.062 ^ab^	0.047 ^a^	0.109 ^c^	***

Different letters in the same row indicate significant differences at the 95% confidence level obtained by the LSD test. *** *p* < 0.001; ns: not significant.

## Data Availability

The original contributions presented in the study are included in the article/Appendix A, further inquiries can be directed to the corresponding author.

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
