# Peer review of "Evaluation of Consumer Perception of New Aquaculture Products through Applying Focus Group and Check-All-That-Apply Methodologies"

_foods, 2024, doi:10.3390/foods13162480_

Round 1

Reviewer 1 Report

Comments and Suggestions for Authors

The topic is timely and innovative. However, the authors should provide detailed information on the decisions made during research. I recommend some reformulations on the introduction, methods, results and discussion & conclusions, before being accepted, according to the following comments. 

Introduction - Why did the authors decide to talk directly on the focus group and not to talk a bit more about co-creation processes and their importance on product development, and then explore the techniques that can be used for this proposal?

Section 2.1.1 - Step 4 and 5 are numbered while the other steps are described in full. Pleas consider to review the paragraph for consistency.

Section 3 - Consider change the figure captions, what is written it seems a sentence not a caption.

It is unclear to me if the ideas generated on the co-creation process were used on the CATA questionnaire. Can you please clarify?

Fig4 - Due to the median value of the variability obtained in CA (59.7%) on the first two factors, did the authors explore other dimensions to see if some other relationship emerged?

Also I would prefer to see directly on the graph the ages. It becomes more clear.

Conclusion:

- Should be improved, given the work done I think it should have a more robust conclusion. 

Comments on the Quality of English Language

English seems good and understandable by non-natives.

Reviewer 2 Report

Comments and Suggestions for Authors

This study aimed to generate ideas for new products derived from different aquaculture species and gain insight into consumer opinions. However, there are some weaknesses that need to be addressed.

1. Abstract: should include more conclusions about the CATA data and correspondence analysis, provide the meaning of the study in the last sentence. In the abstract, why does more analytical method and result data focus on sea bream and prawns? The CATA analysis in the manuscript contained sea bream, lobster, prawn, sea bass, and salmon. The same problem was shown in Key words.

2. Line 106-107: The concepts for novel aquaculture products delineated during the co-creation phase were evaluated across five aquaculture species (sea bream and prawn) and three additional aquaculture species (sea bass, salmon, and lobster) using the CATA methodology.

How to get the five aquaculture species (sea bream and prawn)? Do sea beam and prawn contain different varieties? Why do the CATA analysis data about sea bream and prawn only have one result in Results and Discussion?

Line 106-107, Line 150: How do you define the “novel” or “new” aquaculture products? Not on sale in the market or using new technology?

3. Line 110: suggest to provide the electronic questionnaire used for this study in the supplement materials.

4. 3.1 Focus group: why do the new product ideas results only contain sea bream and prawns, not contain sea bass, prawn and lobster.

5. Line 254: please provide the full name of CA.

6. Line 256: which collectively explained 89.26% of the variability in the data. The explained variance should be 89.25%, not 89.26%.

Reviewer 3 Report

Comments and Suggestions for Authors

Thank you very much for allowing me to review this manuscript titled "Evaluation Of Consumer Perception of New Aquaculture Products Through Applying Focus Group and Check-All-That-Apply Methodologies," Even when the topic could be interesting for the journal readers, there are several elements that do not allow me to accept this manuscript. 

1. The introduction only contains the first paragraph as an actual introduction. The rest should go to the methodology. I suggest adding the fundamental need to have more aquaculture products.

2. The study seems to be multimethod. If this is the case, it must be explained in the text. 

3. the focus groups need to be better explained. I need help understanding how the authors conducted the focus groups or how they analyzed them. For example, it needs to be understood how the authors used the 90 minutes the focus groups lasted. Apparently, what took place was a work workshop, not focus groups. The references used here do not correspond to examples of studies incorporating this methodology in similar studies, as the authors propose (references 16,17).

4. The data analysis of the qualitative information needs to be adequately described. Only the new product options were analyzed numerically, limiting the qualitative methodology. They could have used a survey.

5. Data analysis makes age and gender relevant, but these variables are not described in the introduction or the objective.

6. Conflicts of interest need to be correctly declared. Readers of the journal must identify possible conflicts of interest from authors.

Round 2

Reviewer 2 Report

Comments and Suggestions for Authors

accept in present form.

Reviewer 3 Report

Comments and Suggestions for Authors

Thank to the authors for this new revised version of the manuscript. It has been improved in several aspects that I was concerned about. 

However, I still have some comments. 

- Line 33 to 34. What do the authors mean by whole fish in that context?

- In line 35, the authors state, "Consumers demand aquaculture products." Is there any reference that supports this statement? I need help finding reference 6 as a source to support this idea. 

- Thank you for adding gender and age to the study aim. However, I need help understanding the argument for introducing these variables into the aim. The authors should explain why these variables need to be studied in the introduction. 

-I insist that focus groups and qualitative analysis of them differ from what the authors did. The reference used here describes the regular focus group format, which differs from what the authors did. This methodology is probably a workshop methodology or, at most, discussion sessions. 

- Line 109. What categories were selected, and how were they chosen?

-Line 120. Reference #20 refers to the regular focus group. They used a coding system, and those codes were compared, a technique commonly used in qualitative analysis. It is different from what the authors did in this study. 

- In the discussion, I missed something related to ultra-processed food, what participants co-created, and health.
